# Mildly Constrained Evaluation Policy for Offline Reinforcement Learning

**Linjie Xu**                                                              *linjie.xu@qmul.ac.uk*
*Queen Mary University of London*

**Zhengyao Jiang**                                                         *z.jiang@cs.ucl.ac.uk*
*University College London*

**Jinyu Wang, Lei Song and Jiang Bian**      *{wang.jinyu, lei.song, jiang.bian}@microsoft.com*
*Microsoft Research Asia*

**Reviewed on OpenReview:** *https://openreview.net/forum?id=imAROs79Pb*

## Abstract

Offline reinforcement learning (RL) methodologies enforce constraints on the policy to adhere closely to the behavior policy, thereby stabilizing value learning and mitigating the selection of out-of-distribution (OOD) actions during test time. Conventional approaches apply identical constraints for both value learning and test time inference. However, our findings indicate that the constraints suitable for value estimation may in fact be excessively restrictive for action selection during test time. To address this issue, we propose a *Mildly Constrained Evaluation Policy (MCEP)* for test time inference with a more constrained *target policy* for value estimation. Since the *target policy* has been adopted in various prior approaches, MCEP can be seamlessly integrated with them as a plug-in. We instantiate MCEP based on TD3BC (Fujimoto & Gu, 2021), AWAC (Nair et al., 2020) and DQL (Wang et al., 2023) algorithms. The empirical results on D4RL MuJoCo locomotion, high-dimensional humanoid and a set of 16 robotic manipulation tasks show that the MCEP brought significant performance improvement on classic offline RL methods and can further improve SOTA methods. The codes are open-sourced at `https://github.com/egg-west/MCEP.git`.

## 1 Introduction

Offline reinforcement learning (RL) extracts a policy from data that is pre-collected by unknown policies. This setting does not require interactions with the environment thus it is well-suited for tasks where the interaction is costly or risky. Recently, it has been applied to Natural Language Processing (Snell et al., 2022; Sodhi et al., 2023), e-commerce (Degirmenci & Jones, 2022) and real-world robotics (Kalashnikov et al., 2021; Rafailov et al., 2021; Kumar et al., 2022; Shah et al., 2022; Bhateja et al., 2023) etc. Compared to the standard online setting where the policy gets improved via trial and error, learning with a static offline dataset raises novel challenges. One challenge is the distributional shift between the training data and the data encountered during deployment. To attain stable evaluation performance under the distributional shift, the policy is expected to stay close to the behavior policy. Another challenge is the "extrapolation error" (Fujimoto et al., 2019; Kumar et al., 2019) that indicates value estimate error on unseen state-action pairs or Out-Of-Distribution (OOD) actions. Worsely, this error can be amplified with bootstrapping and cause instability of the training, which is also known as deadly-triad (Van Hasselt et al., 2018). Majorities of model-free approaches tackle these challenges by either constraining the policy to adhere closely to the behavior policy (Wu et al., 2019; Kumar et al., 2019; Fujimoto & Gu, 2021; Wang et al., 2023) or regularising the Q to pessimistic estimation for OOD actions (Kumar et al., 2020; Lyu et al., 2022). In this work, we focus on *policy constraint* methods.

Policy constraint methods minimize the disparity between the policy distribution and the behavior distribution. Meanwhile, the strength of policy constraints introduces a tradeoff between stabilizing value estimates and attaining better inference performance. While various policy constraints have been developed to address this tradeoff, it remains a common problem for them that an excessively constrained policy enables stable value estimate but degrades the evaluation performance (Kumar et al., 2019; Singh et al., 2022; Yu et al., 2023). In particular, the unstable value training may pose the exploding gradient problem. Under this limitation, the valid constraint strengths may not support the best exploitation of the learned value function. In other words, the learned value function may imply a better solution that the overly contained policy fails to learn (See Figure 2 and more details in Section 5.1). To reveal this, we investigate the strength ranges of stable value learning and of better inference performance. However, the investigation into the latter question is impeded by the existing tradeoff, as it requires tuning the constraint without influencing the value learning. To conduct this investigation, we circumvent the tradeoff and seek solutions through the learned value function.

The idea of our approach is inspired by (Czarnecki et al., 2019), which has shed light on the potential of distilling a student policy that improves over the teacher using the teacher's learned value function. Therefore, we propose to derive an extra *evaluation policy* from the value function. The *evaluation policy* does not join the policy evaluation step thus tunning its constraint does not influence value learning. The actor from the actor-critic is now called *target policy* as it is used only to stabilize the value estimation. With the help of *evaluation policy*, we empirically investigate the constraint strengths for 1) stabilizing value learning and 2) better evaluation performance. The results find that a milder constraint improves the evaluation performance but may fall beyond the constraint space of stable value estimation. This finding indicates that the optimal evaluation performance may not be found under the tradeoff, especially when stable value learning is the priority. Therefore, we propose to separate the problems of value learning and evaluation performance, instead of solving them by tradeoff. Consequently, we propose a novel approach of using a *Mildly Constrained Evaluation Policy (MCEP)* derived from the value function to avoid solving the above-mentioned tradeoff and to achieve better evaluation performance. As the *target policy* is commonly used in previous approaches, our MCEP can be integrated with them seamlessly.

The contributions of this work are concluded as following:

- A novel understanding to the policy constraint methods, showing that the learned value function is not well exploited under the tradeoff between value learning and policy performance. With this insight, we propose to separate the problems of value learning and policy performance instead of solving the tradeoff.

- We propose to use an extra *mildly constrained evaluation policy* with a more constrained *target policy*, to achieve better policy performance and stable value learning simultaneously.

- The performance evaluation on D4RL MuJoCo locomotion, high-dimentional humanoid and a set of 16 robotic manipulation tasks show that the MCEP obtains significant performance improvement for policy constraint methods. Moreover, a comparison to inference-time action selection methods and 3 groups of ablation study verifies the significance of the proposed method.

## 2   Related Work

policy constraint method (or behavior-regularized policy method) (Wu et al., 2019; Kumar et al., 2019; Siegel et al., 2020; Fujimoto & Gu, 2021) forces the policy distribution to stay close to the behavior distribution. Different discrepancy measurements such as KL divergence (Jaques et al., 2019; Wu et al., 2019), reverse KL divergence (Cai et al., 2022) and Maximum Mean Discrepancy (Kumar et al., 2019) are applied in previous approaches. (Fujimoto & Gu, 2021) simply adds a behavior-cloning (BC) term to the online RL method Twin Delayed DDPG (TD3) (Fujimoto et al., 2018) and obtains competitive performances in the offline setting. While the above-mentioned methods calculate the divergence from the data, (Wu et al., 2022) estimates the density of the behavior distribution using VAE, and thus the divergence can be directly calculated. Except for explicit policy constraints, implicit constraints are achieved by different approaches. E.g. (Zhou et al.,

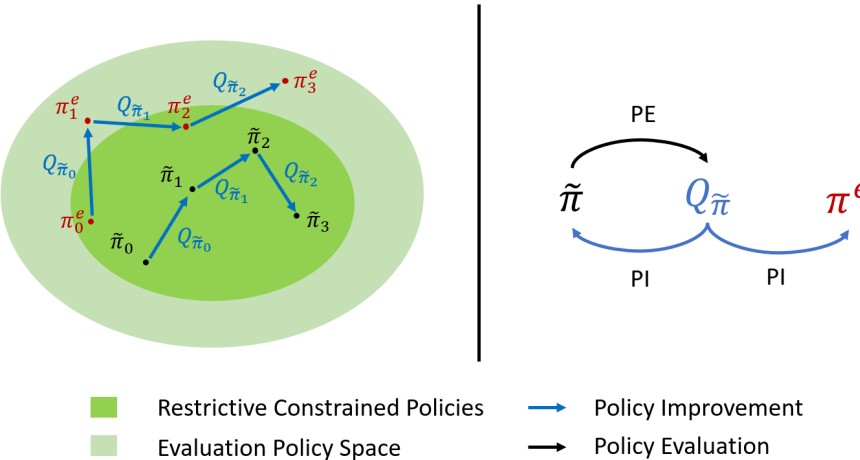

Figure 1: **Left:** diagram depicts policy trajectories for target policy $\tilde{\pi}$ and MCEP $\pi^e$. **Right:** policy evaluation steps to update $Q_{\tilde{\pi}}$ and policy improvement steps to update $\tilde{\pi}$ and $\pi^e$.

2021) ensures the output actions stay in support of the data distribution by using a pre-trained conditional VAE (CVAE) decoder that maps latent actions to the behavior distribution. In all previous approaches, the constraints are applied to the learning policy that is queried during policy evaluation (value learning) and is evaluated in the environment during deployment. Our approach does not count on this learning policy for the deployment, instead, it is used as a *target policy* only for the value learning.

The policy constraint, well-known to be efficient to reduce extrapolation errors, can degrade the policy performance when it is overly restrictive. (Kumar et al., 2019) reveals a tradeoff between reducing errors in the Q estimate and reducing the suboptimality bias that degrades the evaluation policy. A constraint is designed to create a policy space that ensures the resulting policy is under the support of the behavior distribution for mitigating bootstrapping error. (Singh et al., 2022) discussed the inefficiency of policy constraints on *heteroskedastic* dataset where the behavior varies across the state space in a highly non-uniform manner, as the constraint is state-agnostic. A reweighting method is proposed to achieve a state-aware distributional constraint to overcome this problem. Instead of study this well-known trade-off, we propose to separate the problem of value learning and policy performance and devise a solution of circumventing the tradeoff by using an extra *evaluation policy*.

There are methods that extract an evaluation policy from a learned Q estimate. One-step RL (Brandfon-brener et al., 2021; Li et al., 2023) first estimates the behavior policy and its Q estimate, which is later used for extracting the evaluation policy. Although its simplicity, one-step RL is found to perform badly in long-horizon problems due to a lack of iterative dynamic programming (Kostrikov et al., 2022). (Kostrikov et al., 2022) proposed Implicity Q learning (IQL) that avoids query of OOD actions by learning an upper expectile of the state value distribution. No explicit target policy is modeled during their Q learning. With the learned Q estimate, an evaluation policy is extracted using advantage-weighted regression (Wang et al., 2018; Peng et al., 2019). Our approach has a similar form of extracting an evaluation policy from a learned Q estimate. However, one-step RL aims to avoid distribution shift and iterative error exploitation during iterative dynamic programming. IQL avoids error exploitation by eliminating OOD action queries and abandoning policy improvement (i.e. the policy is not trained against the Q estimate). Our work instead tries to address the error exploitation problem and evaluation performance by using policies of different constraint strengths.

## 3 Background

We model the environment as a Markov Decision Process (MDP) $\langle S, A, R, T, p_0(s), \gamma, \rangle$, where $S$ is the state space, $A$ is the action space, $R$ is the reward function, $T(s'|s,a)$ is the transition probability, $p_0(s)$ is initial

state distribution and $\gamma$ is a discount factor. In the offline setting, a static dataset $\mathcal{D}_\beta = \{(s, a, r, s')\}$ is pre-collected by a behavior policy $\pi_\beta$. The goal is to learn a policy $\pi_\phi(s)$ with the dataset $\mathcal{D}$ that maximizes the discounted cumulated rewards in the MDP:

$$\phi^* = \arg\max_\phi \mathbb{E}_{s_0 \sim p_0(\cdot), a_t \sim \pi_\phi(s_t), s_{t+1} \sim T(\cdot|s_t, a_t)} \left[ \sum_{t=0}^{\infty} \gamma^t R(s_t, a_t) \right] \tag{1}$$

Next, we introduce the general policy constraint method, where the policy $\pi_\phi$ and an off-policy Q estimate $Q_\theta$ are updated by iteratively taking policy improvement steps and policy evaluation steps, respectively. The policy evaluation step minimizes the Bellman error:

$$\mathcal{L}_Q(\theta) = \mathbb{E}_{s_t, a_t \sim \mathcal{D}, a_{t+1} \sim \pi_\phi(s_{t+1})} \left[ \left( Q_\theta(s_t, a_t) - (r + \gamma Q_{\theta'}(s_t, a_{t+1})) \right)^2 \right]. \tag{2}$$

where the $\theta'$ is the parameter for a delayed-updated target Q network. The Q value for the next state is calculated with actions $a_{t+1}$ from the learning policy that is updated through the policy improvement step:

$$\mathcal{L}_\pi(\phi) = \mathbb{E}_{s \sim \mathcal{D}, a \sim \pi_\phi(s)} [-Q_\theta(s, a) + wC(\pi_\beta, \pi_\phi)], \tag{3}$$

where $C$ is a constraint measuring the discrepancy between the policy distribution $\pi_\phi$ and the behavior distribution $\pi_\beta$. The $w \in (0, \infty]$ is a weighting factor. Different kinds of constraints were used such as Maximum Mean Discrepancy (MMD), KL divergence, and reverse KL divergence.

## 4 Method

In this section, we first introduce the generic algorithm that can be integrated with any policy constraint method. Next, we introduce three examples based on offline RL methods TD3BC, AWAC and DQL. With a mildly constrained evaluation policy, we name these three instances as *TD3BC-MCEP, AWAC-MCEP* and *DQL-MCEP*.

### 4.1 Offline RL with mildly constrained evaluation policy

The proposed method is designed to overcome the tradeoff between stable value learning and a performant evaluation policy. In previous constrained policy methods, a restrictive policy constraint is applied to obtain stable value learning. We retain this benefit but use this policy (actor) $\tilde{\pi}_\psi$ as a *target policy* only to obtain stable value learning. To achieve better evaluation performance, we introduce an MCEP $\pi_\phi^e$ that is updated by taking policy improvement steps with the value function $Q_{\tilde{\pi}_\psi}$. Different from $\tilde{\pi}_\psi$, $\pi_\phi^e$ does not participate in the policy evaluation procedure. Therefore, a mild policy constraint can be applied, which helps $\pi_\phi^e$ go further away from the behavior distribution without influencing the stability of value learning. We demonstrate the policy spaces and policy trajectories for $\tilde{\pi}_\psi$ and $\pi_\phi^e$ in the l.h.s. diagram of Figure 1, where $\pi_\phi^e$ is updated in the wider policy space using $Q_{\tilde{\pi}_\psi}$.

The overall algorithm is shown as pseudo-codes (Alg. 1). At each step, the $Q_{\tilde{\pi}_\psi}$, $\tilde{\pi}_\psi$ and $\pi_\phi^e$ are updated iteratively. A policy evaluation step updates $Q_{\tilde{\pi}_\psi}$ by minimizing the TD error (line 6), i.e. the deviation between the approximate $Q$ and its target value. Next, a policy improvement step updates $\tilde{\pi}_\psi$ (line 8. These two steps form the actor-critic algorithm. After that, $\pi_\phi^e$ is extracted from the $Q_{\tilde{\pi}_\psi}$, by taking a policy improvement step with a policy constraint that is likely milder than the constraint for $\tilde{\pi}_\psi$ (line 9). Many approaches can be taken to obtain a milder policy constraint. For example, tuning down the weight factor $w^e$ for the policy constraint term or replacing the constraint measurement with

---

**Algorithm 1** MCEP Training

1: **Hyperparameters:**
2: LR $\alpha_Q, \alpha_{\tilde{\pi}}, \alpha_{\pi^E}$, EMA $\eta$, $\tilde{w}$ and $w^e$
3: **Initialize:** $\theta, \theta', \psi$, and $\phi$
4: **for** i=1, 2, ..., N **do**
5: $\quad \mathcal{B}_i \sim \mathcal{D}$
6: $\quad \theta \leftarrow \theta - \alpha_Q \nabla \mathcal{L}_Q(\theta, \mathcal{B}_i)$ (Equation 2)
7: $\quad \theta' \leftarrow (1 - \eta)\theta' + \eta\theta$
8: $\quad \psi \leftarrow \psi - \alpha_{\tilde{\pi}} \nabla \mathcal{L}_{\tilde{\pi}}(\psi; \tilde{w}, \mathcal{B}_i)$ (Equation 3)
9: $\quad \phi \leftarrow \phi - \alpha_{\pi^E} \nabla \mathcal{L}_{\pi^e}(\phi; w^e, \mathcal{B}_i)$ (Equation 3)

---

a less restrictive one. Note that the constraint for $\pi_\phi^e$ is necessary (the constraint term should not be dropped) as the $Q_{\tilde{\pi}_\psi}$ has large approximate errors for state-action pairs that are far from the data distribution.

As the evaluation policy $\pi_\phi^e$ is not involved in the actor-critic updates, one might want to update $\pi_\phi^e$ after the convergence of the $Q_{\tilde{\pi}_\psi}$. An experiment to compare these design options can be found in the Appendix Section A.6. Algorithm 1 that simultaneously updates two policies and these updates (line 8 and 9) can be parallelized to achieve little extra training time based on the base algorithm.

### 4.2 Three Examples: TD3BC-MCEP, AWAC-MCEP and DQL-MCEP

**TD3BC with MCEP** TD3BC takes a minimalist modification on the online RL algorithm TD3. To keep the learned policy to stay close to the behavior distribution, a behavior-cloning term is added to the policy improvement objective. TD3 learns a deterministic policy therefore the behavior cloning is achieved by directly regressing the data actions. For TD3BC-MCEP, the *target policy* $\tilde{\pi}_\psi$ has the same policy improvement objective as TD3BC:

$$\mathcal{L}_{\tilde{\pi}}(\psi) = \mathbb{E}_{(s,a)\sim\mathcal{D}}[-\tilde{\lambda}Q_\theta(s, \tilde{\pi}_\psi(s)) + \big(a - \tilde{\pi}_\psi(s)\big)^2], \tag{4}$$

where the $\tilde{\lambda} = \frac{\tilde{\alpha}}{\frac{1}{N}\sum_{s_i,a_i}|Q_\theta(s_i,a_i)|}$ is a normalizer for Q values with a hyper-parameter $\tilde{\alpha}$: The $Q_\theta$ is updated with the policy evaluation step similar to Eq. 2 using $\tilde{\pi}_\psi$. The MCEP $\pi_\phi^e$ is updated by policy improvement steps with the $Q_{\tilde{\pi}}$ taking part in. The policy improvement objective function for $\pi_\phi^e$ is similar to Eq. 4 but with a higher-value $\alpha^e$ for the Q-value normalizer $\lambda^e$. The final objective for $\pi_\phi^e$ is

$$\mathcal{L}_{\pi^e}(\phi) = \mathbb{E}_{(s,a)\sim\mathcal{D}}[-\lambda^e Q(s, \pi_\phi^e(s)) + \big(a - \pi_\phi^e(s)\big)^2]. \tag{5}$$

**AWAC with MCEP** AWAC (Nair et al., 2020) is an advantage-weighted behavior cloning method. As the target policy imitates the actions from the behavior distribution, it stays close to the behavior distribution during learning. In AWAC-MCEP, the policy evaluation follows the Eq. 2 with the target policy $\tilde{\pi}_\psi$ that updates with the following objective:

$$\mathcal{L}_{\tilde{\pi}}(\psi) = \mathbb{E}_{(s,a)\sim\mathcal{D}}\left[-\exp\bigg(\frac{1}{\tilde{\lambda}}A(s,a)\bigg)\log\tilde{\pi}_\psi(a|s)\right], \tag{6}$$

where the advantage $A(s,a) = Q_\theta(s,a) - Q_\theta(s, \tilde{\pi}_\psi(s))$. This objective function solves an advantage-weighted maximum likelihood. Note that the gradient will not be passed through the advantage term. As this objective has no policy improvement term, we use the original policy improvement with KL divergence as the policy constraint and construct the following policy improvement objective:

$$\mathcal{L}_{\pi^e}(\phi) = \mathbb{E}_{s,a\sim\mathcal{D},\hat{a}\sim\pi^e(\cdot|s)}[-A(s,\hat{a}) + \lambda^e D_{KL}\big(\pi_\beta(\cdot|s)||\pi_\phi^e(\cdot|s)\big)] \tag{7}$$

$$= \mathbb{E}_{s,a\sim\mathcal{D},\hat{a}\sim\pi^e(\cdot|s)}[-A(s,\hat{a}) - \lambda^e \log\pi_\phi^e(a|s)], \tag{8}$$

where the weighting factor $\lambda^e$ is a hyper-parameter. Although the Eq. 6 is derived by solving Eq. 8 in a parametric-policy space, the original problem (Eq. 8) is less restrictive even with $\tilde{\lambda} = \lambda^e$ as the gradient back-propagates through the $-A(s, \pi^e(s))$ term. This difference means that even with a $\lambda^e > \tilde{\lambda}$, the policy constraint for $\pi^e$ could still be more relaxed than the policy constraint for $\tilde{\pi}$.

**DQL with MCEP** Diffusion Q-Learning (Wang et al., 2023) is one of the SOTA offline RL methods that applied a highly expressive conditional diffusion model as the policy to handle multimodal behavior distribution. Its policy improvement step is

$$\mathcal{L}_{\tilde{\pi}}(\psi) = \mathbb{E}_{s\sim\mathcal{D},a\sim\tilde{\pi}}[-\tilde{\lambda}Q(s,a) + C(\pi_\beta, \tilde{\pi})], \tag{9}$$

where $C(\pi_\beta, \tilde{\pi})$ is a behavior cloning term and $\tilde{\lambda}$ is the Q normalizer, similar to TD3BC. The policy improvement step for the evaluation policy has the same manner as the target policy, except for using a different constraint strength.

$$\mathcal{L}_{\pi^E}(\phi) = \mathbb{E}_{s\sim\mathcal{D},a\sim\pi^E}[-\lambda^E Q(s,a) + C(\pi_\beta, \pi^E)]. \tag{10}$$

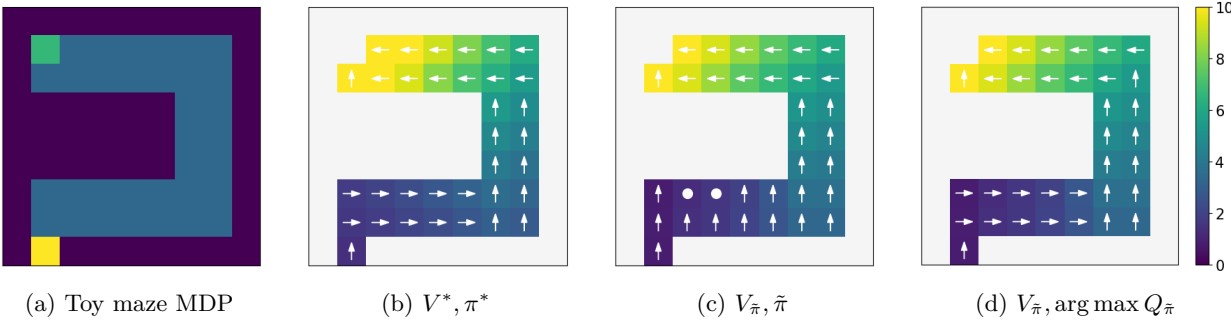

| (a) Toy maze MDP | (b) $V^*, \pi^*$ | (c) $V_{\tilde{\pi}}, \tilde{\pi}$ | (d) $V_{\tilde{\pi}}, \arg\max Q_{\tilde{\pi}}$ |

Figure 2: Evaluation of policy constraint method on a toy maze MDP 2a. In other figures, the color of a grid represents the state value and arrows indicate the actions from the corresponding policy. 2b shows the optimal value function and one optimal policy. 2c shows a constrained policy trained from the above-mentioned offline data, with its value function calculated by $V_\pi = \mathbb{E}_a Q(s, \pi(a|s))$. The policy does not perform well in the low state-value area but its value function is close to the optimal value function. 2d indicates that an optimal policy is recovered by deriving the greedy policy from the off-policy Q estimate (the critic).

## 5 Experiments

In this section, we present experiment results aiming to answer the following research questions. **RQ1.** Does the learned value function imply better solutions than the constrained policy? **RQ2.** Can the solution implied by the value function achievable under current policy constraint methods? **RQ3.** How significantly can the MCEP improve the performance? **RQ4.** How does MCEP perform compared with other action selection methods that also utilize the value function? Additionally, we adopt 2 groups of ablation studies to verify the benefit of an extra *evaluation policy* and *milder constraints*.

**Environments** D4RL (Fu et al., 2020) is an offline RL benchmark consisting of many task sets. Our experiments select 3 versions of **MuJoCo locomotion** (*-v2*) datasets: data collected by rolling out a medium-performance policy (*medium*), the replay buffer during training a medium-performance policy (*medium-replay*), a $50\% - 50\%$ mixture of the medium data and expert demonstrations (*medium-expert*). To investigate more challenging high-dimensional tasks, we additionally collect 3 datasets for **Humanoid-v2** tasks following the same collecting approach of D4RL: *humanoid-medium-v2, humanoid-medium-replay-v2, humanoid-medium-expert-v2*. The humanoid-v2 task has an observation space of 376 dimension and an action space of 17 dimension. This task is not widely used in offline RL research. (Wang et al., 2020; Bhargava et al., 2023) considers this task but our data is independent of theirs. Compared to (Bhargava et al., 2023), we do not consider pure expert data but include the *medium-replay* to study the replay buffer. The statistics of humanoid datasets are listed in Table 5. Finally, we consider a set of 16 complex **Robotic Manipulation** tasks from (Hussing et al., 2023). Their dataset-collecting strategy is similar to the locomotion tasks, hence they are named *manipulation-medium* and *manipulation-medium-replay* in this work.

**Evaluation Protocol** As the offline RL training does not depend on the environment, all the reported results (except for the training process visualization) are produced by evaluating the learned policy on the environment where the data is collected. For visualizing the training process, we save the checkpoints of the policy from different training steps and evaluate them in the environment where the data is collected.

### 5.1 An illustrative example

To investigate the policy constraint under a highly suboptimal dataset, we set up a toy maze MDP that is similar to the one used in (Kostrikov et al., 2022). The environment is depicted in Figure 2a, where the lower left yellow grid is the starting point and the upper left green grid is the terminal state that gives a reward of 10. Other grids give no reward. Dark blue indicates un-walkable areas. The action space is defined as 4 direction movements (arrows) and staying where the agent is (filled circles). There is a 25% probability that

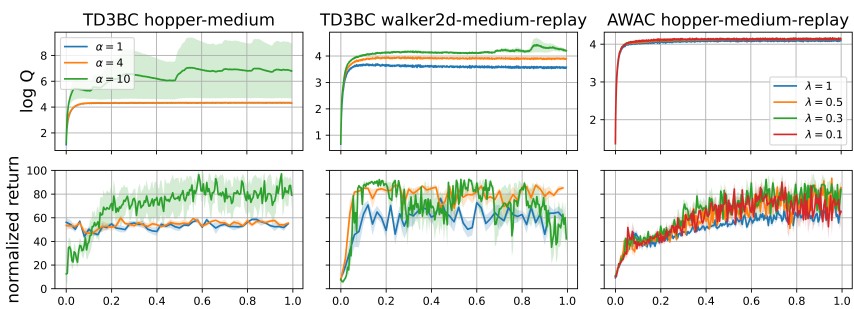 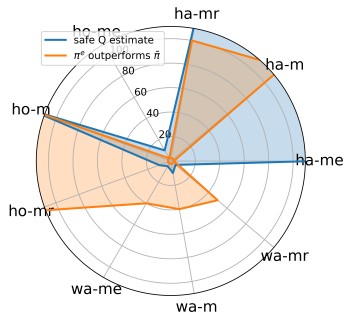

Figure 4: The training process (with standard errors) of TD3BC and AWAC. **Left:** TD3BC on *hopper-medium-v2*. **Middle:** TD3BC on *walker2d-medium-replay-v2*. **Right:** AWAC on *hopper-medium-replay-v2*.

Figure 5: $\alpha$ values in TD3BC for value estimate and test time inference in MuJoCo locomotion tasks.

a random action is taken instead of the action from the agent. For the dataset, 99 trajectories are collected by a uniformly random agent and 1 trajectory is collected by an expert policy. Fig. 2b shows the optimal value function (colors) and one of the optimal policies.

We trained a constrained policy using Eq. 2 and Eq. 8 in an actor-critic manner, where the actor is constrained by a KL divergence with a weight factor of 1. Figure 2c shows the value function and the policy. We observe that the learned value function is close to the optimal one in Figure 2b. However, the policy does not make optimal actions in the lower left areas where the state values are relatively low. As the policy improvement objective shows a trade-off between the Q and the KL divergence, in low-Q-value areas, the KL divergence takes the majority for the learning objective, which makes the policy stay closer to the transitions in low-value areas. However, we find that the corresponding value function indicates an optimal policy. In Figure 2d, we recover a greedy policy underlying the learned value function (Czarnecki et al., 2019) that shows an optimal policy. In conclusion, the constraint might degrade the evaluation performance although the learned value function may indicate a better policy.

## 5.2 Milder constraints potentially improve performance but cause unstable learning

The maze experiment shows that a restrictive constraint might harm the policy performance, which motivates us to deploy milder constraints that potentially better utilize the learned value function. We investigate this question in MuJoCo locomotion tasks. Firstly, we relax the policy constraint on TD3BC and AWAC by setting up different hyper-parameter values that control the strengths of the policy constraints. For TD3BC, we set $\alpha = \{1, 4, 10\}$ (in a descending order of the constraint strengths). For AWAC, we set $\lambda = \{1.0, 0.5, 0.3, 0.1\}$ (in a descending order of the constraint strengths). Finally, We visualize the evaluation performance and the learned Q estimates.

In Figure 4, the left two columns show the training of TD3BC in the *hopper-medium-v2* and *walker2d-medium-replay-v2*. In both domains, we found that using a milder constraint by tuning the $\alpha$ from 1 to 4 improves the evaluation performance, which motivates us to expect better performance with $\alpha = 10$. See from the normalized return of $\alpha = 10$, we do observe higher performances. However, the training is unstable because of the dramatic change in the magnitude of the Q estimates (note the log scale used in the first row). This experiment indicates the tradeoff between the stable Q estimate and the evaluation performance. The rightmost column shows the training of AWAC in *hopper-medium-replay-v2*, we observe higher evaluation performance by relaxing the constraint ($\lambda > 1$). Although the Q estimate keeps stable during the training in all $\lambda$ values, higher $\lambda$ still result in unstable policy performance (the bottom row) and causes the performance crash with $\lambda = 0.1$.

Concluding on all these examples, a milder constraint can potentially improve the performance but may cause unstable Q estimates or unstable policy performances.

| Task Name | BC | | CQL | IQL | EQL | TD3BC | | AWAC | | DQL | |
|---|---|---|---|---|---|---|---|---|---|---|---|
| | 100% | 10% | | | | original | MCEP | original | MCEP | original | MCEP |
| halfcheetah-m | 42.4 | 43.1 | 44.0 | 47.4 | 46.5 | 48.7±0.2 | **55.5±0.4** | 45.1±0 | **46.9±0** | 49.8±0.2 | **53.2±0.2** |
| hopper-m | 54.1 | 56.9 | 58.5 | 65 | 67 | 56.1±1.2 | **91.8±0.9** | 58.9±1.9 | **98.1±0.6** | 81.7±6.6 | **95.5±2.2** |
| walker2d-m | 71 | 73.3 | 72.5 | 80.4 | 81.8 | 85.2±0.9 | **88.8±0.5** | 79.6±1.5 | 81.4±1.6 | 85.5±0.8 | 75.3±3.6 |
| halfcheetah-m-r | 37.8 | 39.9 | 45.5 | 43.2 | 43.1 | 44.8±0.3 | **50.6±0.2** | 43.3±0.1 | **44.9±0.1** | 47±0.2 | **47.8±0.1** |
| hopper-m-r | 22.5 | 72 | 95.0 | 74.2 | 87.9 | 55.2±10.8 | **100.9±0.4** | 64.8±6.2 | **101.1±0.2** | 100.6±0.2 | 100.9±0.3 |
| walker2d-m-r | 14.4 | 56.6 | 77.2 | 62.7 | 71.4 | 50.9±16.1 | **86.3±3.2** | 84.1±0.6 | 83.4±0.8 | 93.6±2.5 | 92.6±2.1 |
| halfcheetah-m-e | 62.3 | 93.5 | 91.6 | 91.2 | 89.4 | 87.1±1.4 | 71.5±3.7 | 77.6±2.6 | 69.5±3.8 | 95.7±0.4 | 93.4±0.8 |
| hopper-m-e | 52.5 | 108.9 | 105.4 | 110.2 | 97.3 | 91.7±10.5 | 80.1±12.7 | 52.4±8.7 | **84.3±16.4** | 102.1±3.0 | **107.7±1.5** |
| walker2d-m-e | 107 | 111.1 | 108.8 | 111.1 | 109.8 | 110.4±0.5 | **111.7±0.3** | 109.5±0.2 | 110.1±0.2 | 109.5±0.1 | 109.7±0.0 |
| Average | 51.5 | 72.8 | 77.6 | 76.1 | 77.1 | 70.0 | **81.9** | 68.3 | **79.9** | 85 | **86.2** |

Table 1: Normalized episode returns on D4RL benchmark. The results (except for CQL) are means and standard errors from the last step of 5 runs using different random seeds. Performances that are higher than corresponding baselines are bolded and task-wise best performances are underlined.

### 5.3 The Evaluation policy allows milder constraints under a stable learning

In this section, we systematically study the constraint strengths on the learning stability and the policy performance. For policy constraint method such as TD3BC, only constraint strengths that do not cause unstable value estimate are valid. To reveal the range of valid strengths, we tune the $\alpha$ for TD3BC within $\mathbb{S} = \{2.5, 5, 10, 20, 30, 40, 50, 60, 70, 80, 90, 100\}$. For each $\alpha$ ($\alpha^e$), we deploy 5 training with different random seeds. In Figure 5, we visualize the unveiled "safe Q estimate" zone, where the constraint strength enables a stable Q estimate for all seeds. The edge of blue area shows the lowest $\alpha$ value that causes Q value explosion. We found that in 4 of the 9 environments, unstable learning doesn't show up with all constraint strength considered. However, in the remaining 5 environments, the valid strengths is relative narrow.

Next, we are interested in the constraint strengths for the policy performance. We investigate it with the help of evaluation policy. We tune the $\alpha^e$ for the *evaluation policy* (TD3BC-EP) within $\mathbb{S}$, with a fixed $\tilde{\alpha} = 2.5$. The orange area in Figure 5 shows the range of $\alpha^e$ where the learned evaluation policy outperforms the target policy. Its edge (the orange line) shows the lowest $\alpha^e$ values where the evaluation policy performance is worse than the target policy.

Note that $\alpha$ weighs the Q term and thus a larger $\alpha$ indicates a less restrictive constraint. Observed from the orange area, we find that in 7 out of the 9 tasks (7 axis where the orange range is not zero), the evaluation policy achieves better performance than the target policy ($\tilde{\alpha} = 2.5$). In 5 tasks (5 axis where the orange range is larger than the blue one), the evaluation policy allows milder policy constraints which cause unsafe q estimate in TD3BC. In conclusion, evaluation policy allows milder policy constraints for potentially better performance and does not influence the Q estimate.

### 5.4 Performance evaluation on MuJoCo locomotion and Robotic Manipulation tasks

**D4RL MuJoCo Locomotion** We compare the proposed method to behavior cloning, classic offline RL baselines AWAC, TD3BC, CQL and IQL, along with SOTA offline RL methods Extreme Q-Learning (EQL) (Garg et al., 2023) and DQL (Wang et al., 2023). Following (Fujimoto & Gu, 2021), each method uses similar hyperparameters for all datasets. The full list of hyper-parameters can be found in Section A.1.

As is shown in Table 1, we observe that the MCEP significantly outperforms their corresponding base algorithm (labeled "original"). TD3BC-MCEP gains significant progress on all *medium* and *medium-replay* datasets. Although the progress is superior, we observe a performance degradation on the *medium-expert* datasets which indicates an overly relaxed constraint for the evaluation policy. Nevertheless, the TD3BC-MCEP achieves a much better average performance than the original algorithm. We also provide a performance comparison between TD3BC and TD3BC-MCEP with their hyperparameters tuned task-wise (Section A.4), where we find that TD3BC-MCEP outperforms TD3BC in 7 of the 9 tasks. In the AWAC-MCEP, we observe a consistent performance improvement over the original algorithm on most tasks and the average performance outperforms the original algorithm significantly. Additionally, evaluation policies

from both TD3BC-MCEP and AWAC-MCEP outperform the CQL, IQL, and EQL while the target policies have relatively mediocre performances. On the SOTA method, DQL, we found that the MCEP obtains further performance improvement although the improvement is not as large as on conventional methods. This difference may caused by an inference-time action selection method DQL uses. i.e. using the learned value function to filter out an action of high approximate value from the policy distribution, which implicitly loose the constraint. We compare the MCEP with inference-time action selection methods in Section 5.5.

**Humanoid** One of the major challenges for offline RL is the distributional shift. In high-dimensional environments, this challenge is exacerbated as the collected data is relatively more limited. To evaluate the proposed method on the ability to handle these environments, we collect 3 datasets from the MoJoCo Humanoid task. Following the naming of D4RL, we name these

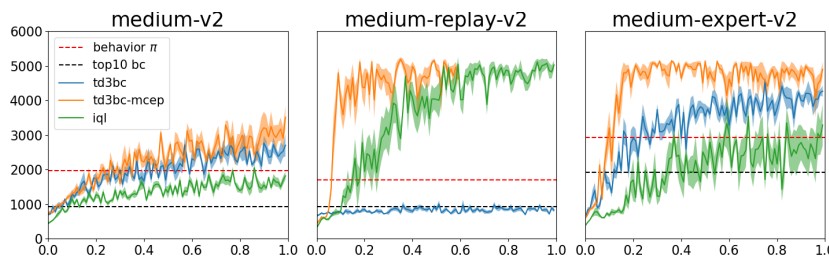

Figure 6: The returns with standard errors during the training on 3 humanoid tasks.

datasets as *medium*, *medium-replay* and *medium-expert*. The details of data collection and the dataset statistics can be found in Section A.2.

We compare the TD3BC-MCEP with BC, TD3BC, CRR Wang et al. (2020), IQL and the behavior policy. As seen in Figure 6, TD3BC-MCEP achieves the highest returns in *medium* and *medium-expert*. Both of these datasets are collected by rolling out the learned online policy. In *medium-replay*, where the dataset is the replay buffer of the online training, TD3BC-MCEP also achieves superior performance and shows a faster convergence rate than IQL. Based on the results, we conclude that the MCEP significantly improves the performance on the original algorithm for high-dimensional environments.

**Robotic Manipulation** Robotic manipulation tasks are recognized as complex tasks for offline RL. We took 16 tasks on the KUKA's IIWA robot from the composition suite (Hussing et al., 2023). These tasks consist of 4 basic tasks *pickplace, push, shelf, trashcan* and 4 target objects *box, dumbbell, hollowbox, plate*. We consider *Medium* and *Medium-Replay* datasets. On these tasks, we compare TD3BC-MCEP with BC, CRR, IQL and TD3BC. Similar to the locomotion setting, we consider similar hyperparameter for all 16 tasks. The hyperparameter details and full results can be found in Section A.3. The overall results are presented in Figure 7, where we observe that while TD3BC fails to compete with other methods in *Medium* and fails to outperform IQL in *Medium-Replay*, the MCEP

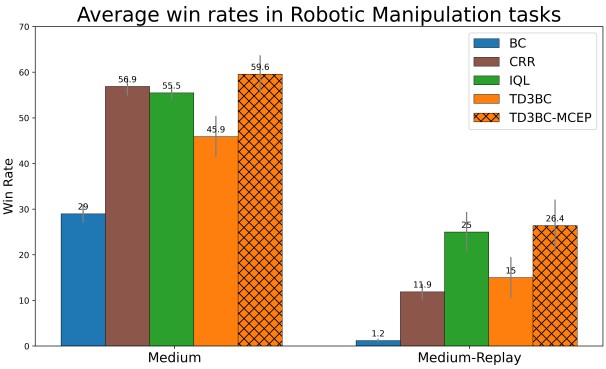

Figure 7: Evaluation (with standard errors) on 16 Robotic Manipulation tasks.

achieves the highest win rates and outperforms all the baselines in both datasets. The results show that the MCEP is able to improve the performance from the base algorithm even in complex domains.

## 5.5 A comparison to Inference-time action selection methods

The inference-time action selection methods (Wang et al., 2020) provides an on-the-fly action selection approach by looking into the value function outputs. As the MCEP also utilizes the learned value function to generate the evaluation policy, one might want to know the performance difference between the MCEP and the inference-time action selection methods. We consider two types of action selection methods. *Argmax*: select the action with the highest estimated q value. *Softmax*: sampling action with the probability proportion to their estimated q values. We compared TD3BC and TD3BC-MCEP on the humanoid tasks. To

generate action samples, we add Gaussian noise to the policy outputs. We consider standard deviation of $[0.01, 0.02, 0.05, 0.1]$ and consider sample size of $[20, 50, 100]$. The best results are visualized in Figure 8.

From Figure 8, we observe that these action selection methods can improve the performance on both TD3BC and TD3BC-MCEP. By comparing the TD3BC with action selection and TD3BC-MCEP without action selection, we note that with *Argmax*, TD3BC outperforms TD3BC-MCEP on *Medium* dataset. However, the *Softmax* does not achieve the same-level performance. In other datasets, even with action selection, TD3BC still fails to compete with TD3BC-MCEP. In conclusion, MCEP brought more significant performance improvement than inference-time action selection methods. The full result can be found in the Appendix A.5.

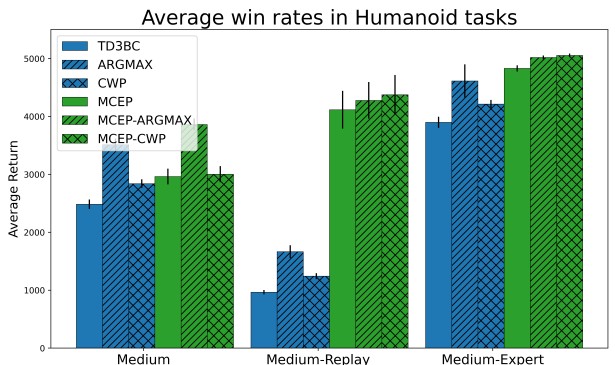

Figure 8: Comparison with Inference-time action-selection methods.

## 5.6 Ablation Study

In this section, we design 2 groups of ablation studies to investigate the effect of the extra evaluation policy and its constraint strengths. Reported results are averaged on 5 random seeds.

**Performance of the extra evaluation policy.** Now, we investigate the performance of the introduced evaluation policy $\pi^e$. For TD3BC, we set the parameter $\alpha = \{2.5, 10.0\}$. A large $\alpha$ indicates a milder constraint. After that, we train TD3BC-MCEP with $\tilde{\alpha} = 2.5$ and $\alpha^e = 10.0$. For AWAC, we trained AWAC with the $\lambda = \{1.0, 0.5\}$ and AWAC-MCEP with $\tilde{\lambda} = 1.0$ and $\lambda^e = 0.5$.

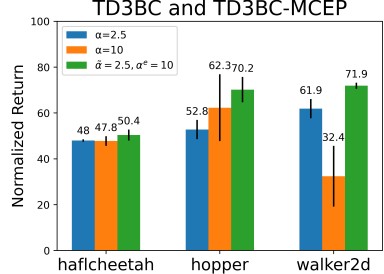 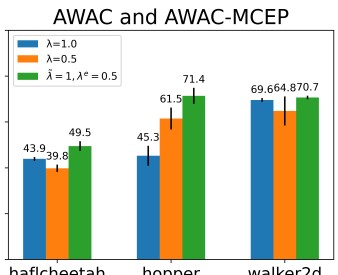

Figure 9: **Left**: TD3BC with $\alpha = 2.5$, $\alpha = 10$ and TD3BC-MCEP with $\tilde{\alpha} = 2.5, \alpha^e = 10$. **Right:** AWAC with $\lambda = 1.0$, $\lambda = 0.5$ and AWAC-MCEP with $\tilde{\lambda} = 1.0$ and $\lambda^e = 0.5$. The standard errors are also plotted.

The results are shown in Figure 9. The scores for different datasets are grouped for each domain. By comparing TD3BC of different $\alpha$ values, we found a milder constraint ($\alpha = 10.0$) brought performance improvement in hopper tasks but degraded the performance in walker2d tasks. The degradation is potentially caused by unstable value estimates (see experiment at section 5.2). Finally, the *evaluation policy* ($\alpha^E = 10.0$) with a *target policy* of $\tilde{\alpha} = 2.5$ achieves the best performance in all three tasks. In AWAC, a lower $\lambda$ value brought policy improvement in hopper tasks but degraded performances in half-cheetah and walker2d tasks. Finally, an evaluation policy obtains the best performances in all tasks.

In conclusion, we observe consistent performance improvement brought by an extra MCEP that circumvents the tradeoff brought by the constraint.

**Constraint strengths of the evaluation policy.** We set up two groups of ablation experiments to investigate the evaluation policy performance under different constraint strengths. For TD3BC-MCEP, we tune the constraint strength by setting the Q normalizer hyper-parameter $\alpha$. The target policy is fixed to $\tilde{\alpha} = 2.5$. We pick three strengths for evaluation policy $\alpha^e = \{1.0, 2.5, 10.0\}$ to create more restrictive, similar, and milder constraints, respectively. For AWAC-MCEP, the target policy has $\tilde{\lambda} = 1.0$. However, it is not straightforward to create a similar constraint for the evaluation policy as it has a different policy improvement objective. We set $\lambda^e = \{0.6, 1.0, 1.4\}$ to show how performance changes with different constraint strengths.

The performance improvements over the target policy are shown in Figure 10. For TD3BC-MCEP, a more restrictive constraint ($\alpha^e = 1.0$) for the evaluation causes a significant performance drop. With a similar constraint ($\tilde{\alpha} = \alpha^e = 2.5$), the performance is slightly improved in two domains. When the evaluation policy has a milder constraint ($\alpha^e = 10$), significant performance improvements are observed in all 3 domains. The right column presents the results of AWAC-MCEP. Generally, the performance in

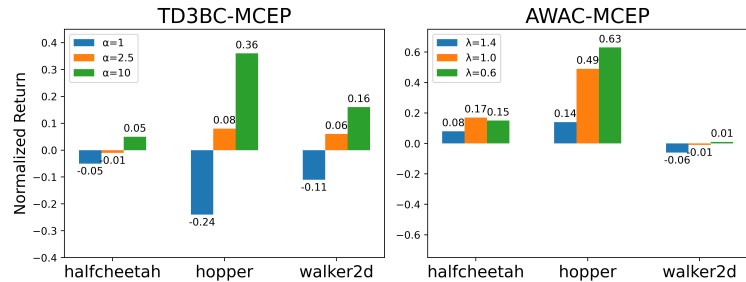

Figure 10: **Left:** TD3BC-EP with $\alpha = 1.0$, $\alpha = 2.5$ and $\alpha = 10.0$. **Right:** AWAC-EP with $\lambda = 1.4$, $\lambda = 1.0$ and $\lambda = 0.6$.

hopper tasks keeps increasing with milder constraints (smaller $\lambda$) while the half-cheetah and walker2d tasks show performances that are enhanced from $\lambda = 1.4$ to $\lambda = 1$ and similar performances between $\lambda = 1$ and $\lambda = 0.6$. It is worth noting that the evaluation policy consistently outperforms the target policy in halfcheetah and hopper domains. On the walker2d task, a strong constraint ($\lambda = 1.4$) causes a performance degradation.

In conclusion, for both algorithms, we observe that on evaluation policy, a milder constraint obtains higher performance than the target policy while a restrictive constraint may harm the performance.

**Estimated Q values for the learned evaluation policies** To compare the performance of the policies on the learning objective (maximizing the Q values), we visualze Q differences between the policy action and the data action $Q(s, \pi(s)) - Q(s, a)$ in the training data (Figure 14, 15 in Section A.7). We find that both the target policy and the MCEP have larger Q estimations than the behavior actions. Additionally, MCEP generally has higher Q values than the target policy, indicating that the MCEP is able to move further toward large Q values.

# 6   Conclusion

This work focuses on the policy constraint methods where the constraint addresses the tradeoff between value estimate and evaluation performance. We first investigate the constraint strength ranges for stable value estimate and for evaluation performance. Our findings indicate that the learned value function is not well exploited under this tradeoff. Then we propose to separate the problems of value learning and policy performance, and devise a simple and general *mildly constrained evaluation policy* approach. The novel approach circumvents the above-mentioned tradeoff thus achieves stable value learning and policy performance simultaneously. The empirical results on locomotion, humanoid and robotic manipulation tasks show that MCEP can obtain significant performance improvement.

**Limitations.** Although the MCEP is able to obtain a better performance, the evaluation policy requires extra effort in tuning its constraint strength. We suggest starting from the strength of the target policy and trying milder constraints. Note that an effective constraint is still indispensable to avoid too many OOD actions that risk real-world application. Moreover, the performance of the MCEP depends on stable value estimation. Unstable value learning may collapse both the target policy and the evaluation policy. While the target policy may recover its performance by iterative policy improvement and policy evaluation (actor-critic), we observe that the evaluation policy may fail to recover its performance from the collapse. Therefore, a restrictive constrained target policy that stabilizes the value learning is essential for the proposed method.

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

# A  Appendix

## A.1  The implementation details and hyper-parameters for locomotion evaluation

For CQL, we reported the results from the IQL paper (Kostrikov et al., 2022) to show CQL results on "-v2" tasks. For IQL, we use the official implementation (Kostrikov, 2022) to obtain a generally similar performance as the ones reported in their paper. Our implementations of TD3BC, TD3BC-MCEP, AWAC, and AWAC-MCEP are based on (Kostrikov, 2022) framework. In all re-implemented/implemented methods, clipped double Q-learning (Fujimoto et al., 2018) is used. In TD3BC and TD3BC-MCEP, we keep the state normalization proposed in (Fujimoto & Gu, 2021) but other algorithms do not use it. For EQL and DQL, we use their official implementation and DQL-MCEP is also built upon the released codebase Wang et al., 2023.

The baseline methods (TD3BC, AWAC and DQL) use the hyper-parameter recommended by their papers. TD3BC uses $\alpha = 2.5$ for its Q value normalizer, AWAC uses 1.0 for the advantage value normalizer and DQL uses $\alpha = 1.0$. In TD3BC-MCEP, the target policy uses $\tilde{\alpha} = 2.5$ and the MCEP uses $\alpha^e = 10$. In AWAC-MCEP, the target policy has $\tilde{\lambda} = 1.0$ and the MCEP has $\lambda^e = 0.6$. In DQL-MCEP, $\tilde{\alpha} = 1.0$ for target policy and $\alpha^e = 2.5$ for evaluation policy. The full list of hyper-parameters used in the experiments can be found in Table 2.

| | BC | IQL | AWAC | AWAC-MCEP | TD3BC | TD3BC-MCEP |
|---|---|---|---|---|---|---|
| actor LR | 1e-3 | 3e-4 | 3e-5 | 3e-5 | 3e-4 | 3e-4 |
| actor^e LR | | - | | 3e-5 | - | 3e-4 |
| critic LR | - | \multicolumn{5}{c}{3e-4} | | | | |
| $V$ LR | - | 3e-4 | \multicolumn{4}{c}{-} | | | |
| actor/critic network | \multicolumn{6}{c}{(256, 256)} | | | | | |
| discount factor | \multicolumn{6}{c}{0.99} | | | | | |
| soft update $\tau$ | - | \multicolumn{5}{c}{0.005} | | | | |
| dropout | 0.1 | \multicolumn{5}{c}{-} | | | | |
| Policy | \multicolumn{4}{c}{TanhNormal} | \multicolumn{2}{c}{Deterministic} | | | |
| \multicolumn{7}{c}{MuJoCo Locomotion} | | | | | | |
| $\tau$ for IQL | - | 0.7 | \multicolumn{4}{c}{-} | | | |
| $\lambda/\tilde{\lambda}$ | - | $1/\lambda = 3$ | 1.0 | \multicolumn{3}{c}{-} | | |
| $\lambda^e$ | \multicolumn{2}{c}{-} | 0.6 | \multicolumn{3}{c}{-} | | |
| $\alpha/\tilde{\alpha}$ | \multicolumn{4}{c}{-} | \multicolumn{2}{c}{2.5} | | |
| $\alpha^e$ | \multicolumn{5}{c}{-} | 10.0 | | |

Table 2: The hyper-parameters for MuJoCo locomotion tasks.

| Task | # of trajectories | # of samples | Mean of Returns |
|---|---|---|---|
| Humanoid Medium | 2488 | 1M | 1972.8 |
| Humanoid Medium Replay | 3008 | 0.502M | 830.2 |
| Humanoid Medium Expert | 3357 | 1.99M | 2920.5 |

Table 3: Dataset statistics for humanoid offline data.

## A.2  Data collection and hyper-parameters tunning for humanoid tasks

**Hyperparameters.** In this experiment, we select Top-10 Behavior cloning, TD3BC and IQL as our baselines. For Top-10 Behavior cloning, only 10% data of highest returns are selected for learning. For TD3BC, we searched the hyperparameter $\alpha = \{0.1, 0.5, 1.0, 2.0, 3.0, 4.0, 5.0\}$. For IQL, we searched the expectile hyperparameter $\tau = \{0.6, 0.7, 0.8, 0.9\}$ and the policy extraction hyperparameter $\lambda = \{0.1, 1.0, 2.0, 3.0\}$. For CRR, we tune the advantage coefficiency $\beta = \{0.1, 0.6, 0.8, 1.0, 1.2, 5.0\}$. For TD3BC-MCEP, we searched the $\tilde{\alpha} = \{0.1, 0.5, 1.0, 2.0, 3.0\}$ and $\alpha^E = \{3.0, 4.0, 5.0, 10.0\}$. The final selected hyperparameters are listed in

Table 4. For CRR, we implement the CRR exp version based on (Hoffman et al., 2020). This version is considered as it outperforms other baselines in (Wang et al., 2020) in complex environments such as humanoid. We also applied *Critic Weighted Policy* as well as an argmax version of it (CRR-argmax). These design options result in CRR, CRR-CWP and CRR-Argmax variants. In Figure 6, we report the most performant CRR variant for each task. Among all its variants, CRR-Argmax shows better performance in both the *medium* and the *medium-replay* while CRR performs the best in the *medium-expert* task.

**Humanoide Data Collection.** In the Table 5, we provide statistics of the collected data.

| | BC | IQL | TD3BC | TD3BC-MCEP |
|---|---|---|---|---|
| actor LR | 1e-3 | 3e-4 | 3e-4 | 3e-4 |
| actor^e LR | | - | - | 3e-4 |
| critic LR | - | 3e-4 | | |
| $V$ LR | - | 3e-4 | - | |
| actor/critic network | (256, 256) | | | |
| discount factor | 0.99 | | | |
| soft update $\tau$ | - | 0.005 | | |
| dropout | 0.1 | - | | |
| Policy | TanhNormal | Deterministic | | |
| Humanoid-medium-v2 | | | | |
| $\tau$ for IQL | - | 0.6 | - | |
| $\lambda/\tilde{\lambda}$ | - | 1 | - | |
| $\alpha/\tilde{\alpha}$ | - | | 1 | 0.5 |
| $\alpha^e$ | - | | | 3 |
| Humanoid-medium-replay-v2 | | | | |
| $\tau$ for IQL | - | 0.6 | - | |
| $\lambda/\tilde{\lambda}$ | - | 0.1 | - | |
| $\alpha/\tilde{\alpha}$ | - | | 0.5 | 1.0 |
| $\alpha^e$ | - | | | 10 |
| Humanoid-medium-expert-v2 | | | | |
| $\tau$ for IQL | - | 0.6 | - | |
| $\lambda/\tilde{\lambda}$ | - | 0.1 | - | |
| $\alpha/\tilde{\alpha}$ | - | | 2 | 0.5 |
| $\alpha^e$ | - | | | 3 |

Table 4: The hyper-parameters for Humanoid task.

### A.3 The full results for the robotics manipulation experimetns

We consider similar hyperparameter for all tasks. To obtain a fair comparison, we extensively search the hyperparameters for all baselines. The CRR is tuned among $\lambda = [0.4, 0.6, 0.8, 1.0, 1.2]$ (1.0 selected) and uses the critic weighted policy (Wang et al., 2020). For TD3BC, we tune $\lambda = [1.0, 2.0, 3.0, 4.0]$ and select 2.0. IQL ueses $\tau = 0.7, \lambda = 3.0$ as recommended in the dataset paper (Hussing et al., 2023). For TD3BC-MCEP, we use $\tilde{\alpha} = 2, \alpha^E = [4.0, 6.0, 8.0, 10.0]$ (8.0 selected).

### A.4 An comparison with task-specific hyper-parameters on locomotion tasks

To investigate the task-specific optimal policy constraint strengths, we search this hyperparameter for TD3BC and TD3BC-MCEP (with $\tilde{\alpha} = 2.5$) in each task of the locomotion set. Their optimal values and the corresponding performance improvement are visualized in Figure 11. As we observed, in 7 of the 9 tasks, the optimal policies found by TD3-MCEP outperform optimal policies found by TD3BC. In all *medium* tasks, though the optimal constraint strenths are the same for TD3BC and TD3BC-MCEP, TD3BC-MCEP outperformance TD3BC. This is benefitted by that relaxing the constraint of evaluation policy does not influence the value estimate. However, for TD3BC, milder constraint might cause unstable value estimate

| Dataset | BC | CRR | IQL | TD3BC | -MCEP |
|---------|----|----|----|------|------|
| | | | Medium | | |
| Box-PickPlace | 10.8(1.3) | 92.8(0.5) | 93.8(2.7) | 89.8(2.9) | **100**(0) |
| Box-Push | 74.6(1.9) | 39.2(4.2) | 91.8(1.1) | 93.8(1.6) | **99.8**(0.2) |
| Box-Shelf | 91.8(1.2) | 91.6(0.7)) | 98.6(0.9) | 93.2(2.8) | **99.2**(0.7) |
| Box-Trashcan | 8.6(2.3) | 24(2.9) | 0(0) | 1.2(1.1) | 0(0) |
| Dumbbell-PickPlace | 38.6(2.7) | 52.2(1.2) | 86.8(2.0) | 63.2(3.0) | **70.4**(9.6) |
| Dumbbell-Push | 55.2(3.7) | 20.6(1.2) | 66.6(2.4) | 54.0(8.6) | **58.0**(10.4) |
| Dumbbell-Shelf | 40.8(4.9) | 50.2(1.5) | 0.6(0.4) | 21.0(4.3) | **44.6**(11.2) |
| Dumbbell-Trashcan | 5.2(0.7) | 62(1.8) | 87.1(3.8) | 28.0(10.2) | **68.2**(16.1) |
| Hollowbox-PickPlace | 42.4(3.5) | 85.8(2.0) | 95.2(2.4) | 82.6(11.1) | **92.2**(3.6) |
| Hollowbox-Push | 0(0) | 55.2(4.8) | 69.4(7.5) | 49.2(4.8) | **98.2**(1.0) |
| Hollowbox-Shelf | 72.2(1.8) | 94(0.7) | 98.2(1.4) | 95.4(1.7) | **98.4**(0.6) |
| Hollowbox-Trashcan | 0(0) | 29.2(2.1) | 0(0) | 0(0) | 0(0) |
| Plate-PickPlace | 0(0.2) | 60.8(3.4) | 1(0.3) | 2.2(0.4) | 0.4(0.2) |
| Plate-Push | 0(0) | 12.6(1.1) | 0(0) | 0(0) | **25.0**(11.9) |
| Plate-Shelf | 24.2(7.4) | 67(2.0) | 99(0.4) | 60.4(19.6) | **99.8**(0.2) |
| Plate-Trashcan | 0.2(0.2) | 74(1.1) | 0.4(0.2) | 0.8(0.2) | 0(0) |
| **Average** | 29.0 | 56.9 | 55.5 | 45.9 | **59.6** |
| | | | Medium-Replay | | |
| Box-PickPlace | 0(0) | 41.4(2.9) | 50.8(11.5) | 23.0(12.1) | 0(0) |
| Box-Push | 0(0) | 2.8(0.8) | 0(0) | 60.8(5.1) | 41.6(10.3) |
| Box-Shelf | 0(0) | 3.0(1.4) | 16.6(4.6) | 6.4(2.1) | **49.8**(14.6) |
| Box-Trashcan | 0(0) | 1.4(0.5) | 92.3(2.6) | 0(0) | **76.2**(6.9) |
| Dumbbell-PickPlace | 0(0) | 25.6(4.6) | 34.1(2.7) | 8.2(6.9) | 0(0) |
| Dumbbell-Push | 4.1(2.1) | 1.4(0) | 3.2(1.6) | 24.2(5.0) | **55.8**(6.0) |
| Dumbbell-Shelf | 9.8(2.3) | 2.6(0.8) | 11.6(6.1) | 25.4(7.8) | 12.0(10.1) |
| Dumbbell-Trashcan | 5.0(2.3) | 17.0(4.3) | 65.0(10.2) | 29.0(5.6) | **95.2**(0.5) |
| Hollowbox-PickPlace | 0(0) | 5.6(3.3) | 0(0) | 6.6(5.9) | 0.4(0.4) |
| Hollowbox-Push | 0(0) | 18.2(3.0) | 30.0(3.7) | 23.0(12.2) | 3.0(2.7) |
| Hollowbox-Shelf | 0(0) | 1.8(0.9) | 61.4(4.7) | 32.0(6.5) | **58.4**(12.6) |
| Hollowbox-Trashcan | 0(0) | 4.4(0.8) | 4.8(4.6) | 0(0) | 0(0) |
| Plate-PickPlace | 0(0) | 31.4(2.2) | 29.4(17.8) | 2.2(2.0) | 0(0) |
| Plate-Push | 0(0) | 0(0) | 0(0) | 0(0) | **10.6**(9.5) |
| Plate-Shelf | 0(0) | 0.8(0.4) | 0(0) | 0(0) | **19.4**(17.4) |
| Plate-Trashcan | 0.8(0.4) | 32.8(1.1) | 1.0(0.3) | 0.4(0.2) | 0(0) |
| **Average** | 1.2 | 11.9 | 25.0 | 15.0 | **26.4** |

Table 5: Win rates with standard errors for Robotic Manipulation tasks. All results are averaged among 5 random seeds. The win rates of the MCEP that outperform the base algorithm are bolded.

during training. In all *medium-replay* tasks, we found optimal constraints for TD3BC-MCEP are milder than TD3BC, which verifies the requirements of milder constraints 5.2.

### A.5 An investigation of other methods for inference-time action selection

The MCEP aims to improve the inference time performance without increasing the Bellman estimate error. Previous works also propose to use the on-the-fly inference-time action selection methods. For example, (Wang et al., 2020) proposes the *Critic Weighted Policy (CWP)*, where the critic is used to construct a categorical distribution for inference-time action selection. Another simple method is selecting the action of the largest Q values, namely **Argmax**. In this section, we compare the performance of TD3BC and TD3BC-MCEP under different test-time action selection methods in the hidh-dimensional Humanoid tasks.

The results are presented in Table 6 and 7. Both the Argmax and the CWP methods select an action from an action set. We generate this action set by adding Gaussian noise to the outputs of the deterministic policy. The *std* is the noise scale and N is the size of this action set. From the results, we observe that

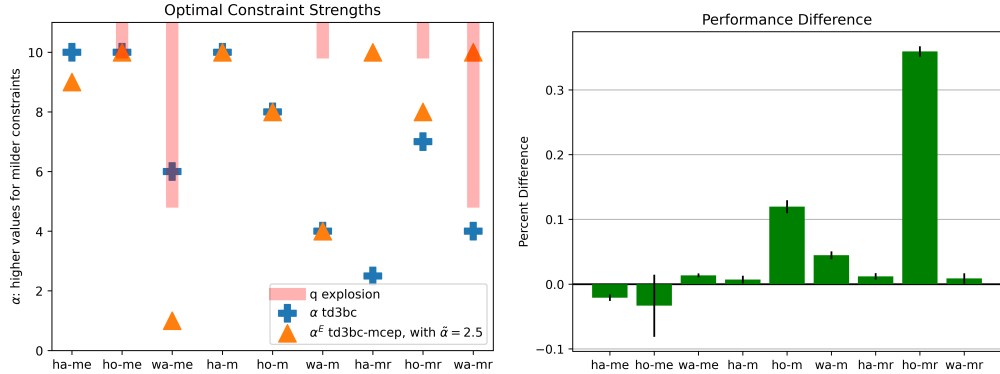

Figure 11: **Left:** Optimal $\alpha^E$ values for the evaluation policy of TD3BC-MCEP, with a fixed $\alpha = 2.5$ for the target policy. Optimal $\tilde{\alpha}$ values for TD3BC. Red areas indicate the $\alpha$ values for TD3BC that raise Q-value explosion (in one or more training of a 5-seed training). **Right:** Performance difference (with standard errors) between the evaluation policy of TD3BC-MCEP and the actor of TD3BC, using the $\alpha^E$ ($\alpha$) values shown in the left figure.

CWP and Argmax help improve the performance of both the TD3BC and TD3BC-MCEP. It is worth noting that, in *medium* task, the Argmax method improves the TD3BC to the same level as TD3BC-MCEP. But in *meidum-replay* and *medium-expert* tasks, the improved performances are still worse than the original TD3BC-MCEP (without using action selection). On TD3BC-MCEP, applying Argmax and CWP further improves policy performances.

In conclusion, the inference-time performance could be improved by utilizing the inference-time action selection methods but MCEP shows a more significant policy improvement and does not show conflict with these action selection methods.

Table 6: TD3BC with inference-time action selection. The original policy has returns 2483.9, 965.4 and 3898.2 for medium, medium-replay and medium-expert, respectively. Standard errors are also reported.

| Game | N\std | Argmax | | | | CWP | | | |
|---|---|---|---|---|---|---|---|---|---|
| | | 0.01 | 0.02 | 0.05 | 0.1 | 0.01 | 0.02 | 0.05 | 0.1 |
| medium | 20 | 2462.4(186.4) | 2944.5(371.3) | 3098.9(149.2) | **3511.1**(244.7) | 2441.4(217.4) | 2689.8(323.2) | 2755.4(303.9) | 3113.2(416.5) |
| | 50 | 2564.2(180.1) | 2836.0(399.7) | 2956.6(128.2) | 3156.3(272.4) | 2159.8(239.8) | 2588.6(181.7) | 2462.1(180.5) | **2839.6(165.6)** |
| | 100 | 2857.0(96.7) | 2369.8(250.4) | 3122.0(228.7) | 3266.8(314.4) | 2607.6(140.9) | 2665.8(335.5) | 2722.5(305.6) | 2584.1(216.7) |
| medium-replay | 20 | 895.3(44.1) | 1042.3(133.3) | 1136.7(135.8) | 1524.1(125.3) | 973.9(75.7) | 931.9(61.9) | 932.2(42.4) | **1242.7(116.5)** |
| | 50 | 994.6(60.3) | 976.7(48.9) | 1160.2(72.8) | **1664.9(248.8)** | 974.7(66.5) | 1030.3(96.5) | 1002.1(87.7) | 1171.1(124.6) |
| | 100 | 971.7(69.5) | 1049.0(55.6) | 1232.2(144.2) | 1574.7(179.9) | 874.2(41.9) | 1023.5(85.2) | 973.0(74.8) | 1232.9(117.2) |
| medium-expert | 20 | 3861.7(345.8) | 4068.4(175.7) | 4131.0(299.7) | 4585.3(206.6) | 4181.1(255.3) | 4478.3(174.2) | 3904.0(166.1) | 3636.7(253.4) |
| | 50 | 4460.6(135.7) | 4012.2(318.7) | **4612.9**(127.7) | 4603.0(137.5) | 3987.0(288.1) | 4068.9(206.0) | 3995.1(323.6) | **4214.7**(157.0) |
| | 100 | 4130.8(301.2) | 4141.7(248.4) | 4158.3(302.8) | 4421.4(130.4) | 4145.3(262.0) | 3634.6(389.4) | 3933.9(249.4) | 3788.3(246.1) |

Table 7: TD3BC-MCEP with inference-time action selection. The original policy has returns 2962.8, 4115.6 and 4829.2 for medium, medium-replay and medium-expert, respectively. Standard errors are also reported.

| Game | N\std | Argmax | | | | CWP | | | |
|---|---|---|---|---|---|---|---|---|---|
| | | 0.01 | 0.02 | 0.05 | 0.1 | 0.01 | 0.02 | 0.05 | 0.1 |
| medium | 20 | 2368.2(221.2) | 2871.4(334.1) | 2924.1(211.8) | 3392.8(319.4) | 2670.5(262.8) | 2710.0(96.8) | 3146.5(216.6) | 2987.9(244.0) |
| | 50 | 2822.1(194.0) | 3046.3(287.2) | 3283.9(298.3) | **3861.7**(225.2) | 2612.9(142.0) | 2787.1(154.9) | 2718.9(344.9) | 2841.7(248.7) |
| | 100 | 3405.1(326.9) | 2808.2(191.8) | 3264.5(310.4) | 3751.3(490.7) | **3003.3**(312.1) | 2896.8(192.2) | 2748.9(163.1) | 2727.2(284.9) |
| medium-replay | 20 | **4277.3**(708.5) | 4071.5(703.4) | 4092.7(694.1) | 4253.4(659.7) | 4033.1(689.8) | 4200.0(652.1) | 4254.4(703.6) | 4167.0(632.7) |
| | 50 | 4225.5(691.8) | 4159.7(681.4) | 4028.4(621.6) | 4210.8(710.2) | 4135.5(651.7) | 4219.1(680.6) | **4375.7**(760.6) | 4230.3(629.4) |
| | 100 | 4190.3(691.4) | 3966.4(639.9) | 4138.4(694.1) | 4270.1(695.0) | 4328.5(733.3) | 4266.9(638.9) | 4275.7(671.3) | 4142.5(598.7) |
| medium-expert | 20 | 4752.8(232.4) | 4956.7(92.6) | 4880.3(229.6) | 4887.1(238.3) | 4736.4(266.5) | 4710.8(226.7) | 4748.7(177.5) | 4942.1(163.0) |
| | 50 | 4930.7(157.6) | **5018.2**(73.6) | 4614.2(150.0) | 4899.9(158.7) | **5053.4**(70.6) | 5001.4(85.0) | 4808.8(183.4) | 4670.6(204.2) |
| | 100 | 4616.8(217.9) | 4800.9(145.4) | 4700.9(179.7) | 4648.0(288.5) | 4588.1(237.3) | 4770.6(86.3) | 4934.3(176.6) | 4855.3(146.9) |

### A.6 The design option of how the evaluation policy update

As the evaluation policy is not involved in the actor-critic's iterative update, one might want to update the evaluation policy after the actor-critic converges, namely **afterward updates**. While this is a valid design option, our method simultaneously updates the target policy and the evaluation policy (**simultaneous updates**). In this manner, their updates can be parallelized and no further time is required based on the actor-critic training. This parallelization can significantly reduce the training time for methods of slow policy update (e.g. DQL). Figure 12 and 13 present the convergence for these two design options. From the results, we observe a faster convergence of afterward updates in some tasks. However, there are also many tasks where the afterward updates method converges after a million steps.

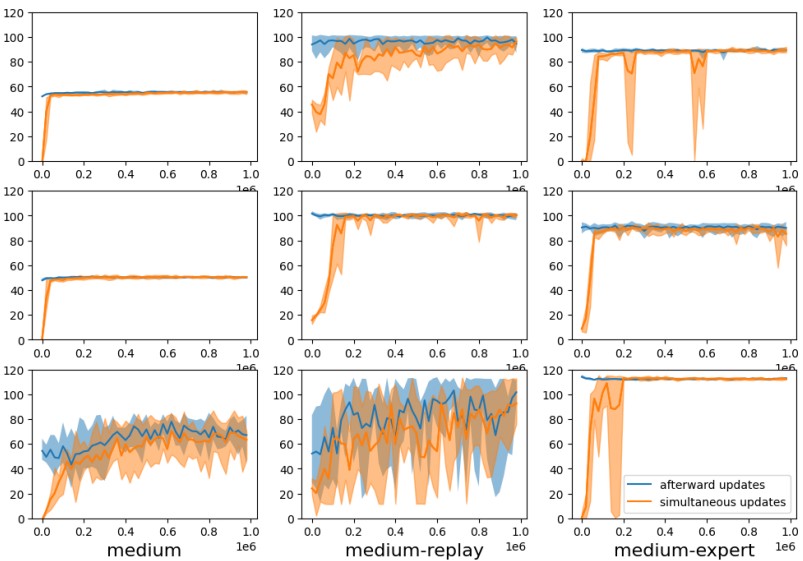

Figure 12: Episode returns with standard errors of *simultaneous updates* and *afterward updates* for the evaluation for TD3BC-MCEP. **First row:** *halfcheetah.* **Second row** *hopper.* **Third row:** *walker2d.*

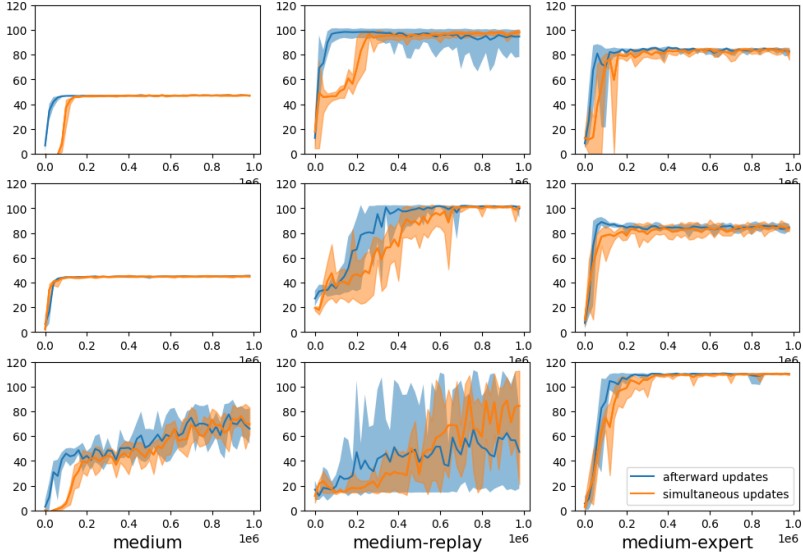

Figure 13: Episode returns with standard errors of *simultaneous updates* and *afterward updates* for the evaluation for AWAC-MCEP. **First row:** *halfcheetah.* **Second row** *hopper.* **Third row:** *walker2d.*

### A.7 The full results for estimated Q values of the learned evaluation policies

Figure 14 and Figure 15 show the visualization of the estimated Q values achieved by the target policy and evaluation policy.

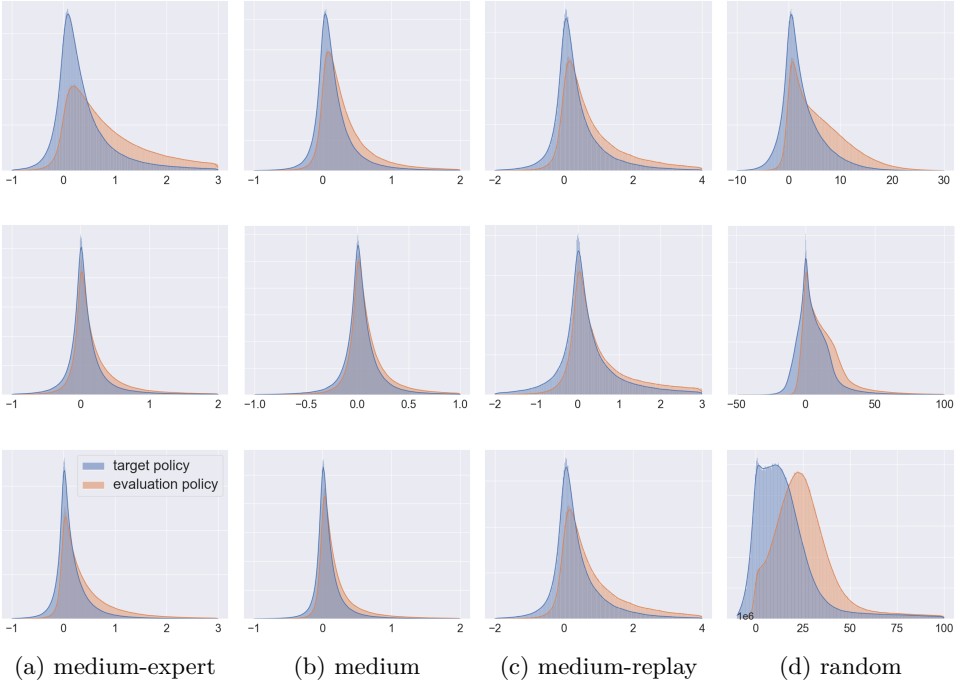

(a) medium-expert     (b) medium     (c) medium-replay     (d) random

Figure 14: TD3BC-MCEP. **First row:** *halfcheetah.* **Second row** *hopper.* **Third row:** *walker2d.*

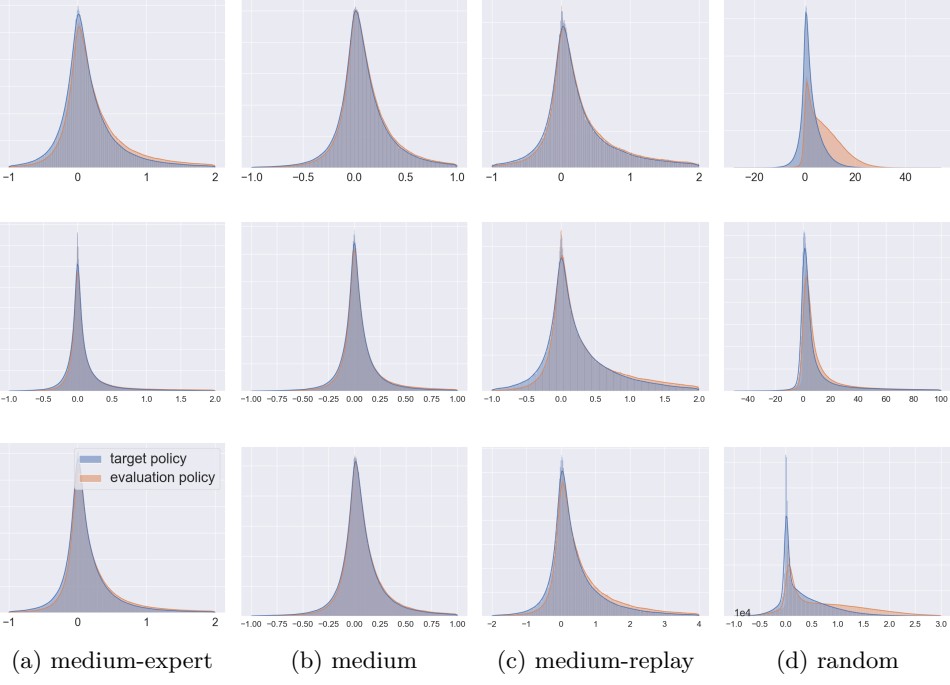

(a) medium-expert     (b) medium     (c) medium-replay     (d) random

Figure 15: AWAC-MCEP. **First row:** *halfcheetah.* **Second row:** *hopper.* **Third row:** *walker2d.*

