# OpenReview forum: "Mildly Constrained Evaluation Policy for Offline Reinforcement Learning"
_TMLR — Accepted by TMLR_

### Review · Reviewer_Wivj · 2024-01-30

**Summary Of Contributions:**

The paper proposes an offline reinforcement learning (RL) approach that separates value learning from policy evaluation. This separation enables the learning of a better policy in the latter stage without affecting the stability of the former stage. To motivate their approach, the authors empirically demonstrate that the critic learned from standard offline RL suggests a stronger policy than the one learned directly. Results on an offline RL benchmark, D4RL, showcase the effectiveness of this approach.

**Audience:**

Yes

**Broader Impact Concerns:**

Authors can potentially discuss the cost of tuning the hyper-parameters and how it effects the safety of this method in real-world applications.

**Claims And Evidence:**

Yes

**Requested Changes:**

- The second and third paragraphs require revision for clarity. Currently, there are two main issues:
  *  The language used in these paragraphs doesn't effectively connect with the experimental questions in section 5. It's initially unclear why those questions are posed. To improve this, try to avoid alternating between terms like "critic" and "value function", especially since they refer to the same concept in this paper's context. Consistently using one term could make the connection more apparent.
  * The necessity of training stability, despite good evaluation performance, isn't immediately evident. If training instability poses a safety risk or if steady improvement is crucial for optimal final performance, this should be explicitly stated. The discussion in the experiment section, which suggests the critic implies a better policy than the learned one, offers a compelling motivation. This point could be introduced earlier to engage readers sooner.
  * Consider integrating the last four sentences of the second paragraph into the third paragraph, where your method is introduced. These sentences, which currently seem to pertain only to evaluation, actually provide valuable insights into the approach's details and benefits. Their current placement might imply they are relevant only in the evaluation context, while the discussion of the proposed approach itself is somewhat brief.

- Introducing a reference to Figure 2 earlier in the paper, possibly in the introduction or just after section 3, could help readers better grasp the issue your research addresses.

- Algorithm 1 needs more precision. Specify whether each update computes loss on the entire dataset or merely a batch of data. Clarify if all models are updated in each iteration with the same batch and if a batch parameter should be included in all the loss functions. Additionally, confirm if all learning rates are identical.

- Please clarify how training and evaluation conditions differ. Are the tests conducted in the training environments, or are the testing environments sampled from the same distribution as the training environments?

- Figure 4 is confusing due to its non-informative caption. Clarify whether these results pertain to training or testing environments. The term "divergence" in value estimate needs definition: does it refer to a high variance in the Q function or a divergence in the curves of different hyper-parameter values? If the variance indicates instability, consider showcasing only the variance over time. Clearly indicate upfront what signs of instability (e.g., larger variance) readers should look for. Additionally, it's unclear why the evaluation performance isn't plotted, especially since it's mentioned in the text. Clarify whether the second row shows training or evaluation performance. The term "evaluation performance" itself is somewhat vague; specify whether it refers to performance in test or training environments.

- In all tables and figures where confidence intervals are reported or plotted, please specify the type of intervals used (e.g., standard error, X% CI derived from standard error, etc.).

**Strengths And Weaknesses:**

Strengths:

* The approach is simple, yet general and effective.
* Results demonstrate that it surpasses various strong baselines in the tasks it was tested on.
* The paper is mostly well-written, and I appreciate how the authors carefully explain their motivation, supporting it with empirical evidence.

Weaknesses:

* The approach requires increased effort in hyper-parameter tuning, which could be challenging when deploying in real-world scenarios where simulators are unavailable.
* The introduction is somewhat obscure and fails to effectively convey the significance of the contributions.
* Some of the plots and their captions are difficult to understand.
* Additionally, several parts of the paper could be reorganized to improve its overall flow.

---

### Review · Reviewer_upJw · 2024-02-12

**Summary Of Contributions:**

This paper studies the policy constraint methods from a novel aspect by showing that the learned value function is not well exploited under the tradeoff between value learning and policy performance. To handling this, it proposes use an extra mildly constrained evaluation policy with a more constrained target policy, to achieve better policy performance and stable value learning simultaneously.

The empirical evaluation on D4RL MuJoCo locomotion, high-dimentional humanoid and a set of 16 robotic manipulation tasks show that the MCEP obtains significant performance improvement for policy constraints methods. Related ablation study is also conducted to for the study.

**Audience:**

Yes

**Broader Impact Concerns:**

This is a study that does not raise Broader Impact Concerns.

**Claims And Evidence:**

Yes

**Requested Changes:**

Please answer my questions in above.

**Strengths And Weaknesses:**

My main concern is that I don't understand how MCEP training is different from the standard actor-critic method. In particular, why line 8 of Algorithm 1 is useful? For instance, the information in line 8 is not leveraged since the computation in Line 9 or Line 6 does not use any information related to psi. The algorithm will run seamlessly without Line 8. In this sense, I don't understand where the improvement from Table 1 comes from.

Second, is there a principled way to compute the hyperparameter lambda so the algorithm can be better than the standard actor-critic? In addition, how's other hyperparameter tuned?

Lastly, it would be nicer if the study could provide some sample complexity guarantees.

Small questions:

1. In Algorithm 1, it should be gradient rather than loss itself in the update;
2. The one-step method [1] seems related, the author should include it and discuss the pros and cons (compared to the current work).
[1] Offline Reinforcement Learning with Closed-Form Policy Improvement Operators, ICML23.

---

### Review · Reviewer_5tm2 · 2024-02-13

**Summary Of Contributions:**

I am honored to review this paper again and I am Reviewer VtJK of this paper at ICLR 2024. I summarize the contributions of this paper below:

In policy constraint offline reinforcement learning (RL) algorithms, it is a common practice that the constraints for both value learning and test time inference are the same. This paper argues that such a paradigm may hinder the performance of the agent during test time inference. To address this issue, they propose the Mildly Constrained Evaluation Policy (MCEP) for test time inference. The idea is quite simple and the implementation is also easy. MCEP has the same objective function as the policy trained during the offline phase, but it does not participate in the policy evaluation phase. The authors show that by doing so, the performance of the offline RL agents can be improved.

**Audience:**

Yes

**Broader Impact Concerns:**

I believe no broader impact concerns are needed to be specified.

**Claims And Evidence:**

Yes

**Requested Changes:**

- Please add error bars in Figure 7, Figure 8, Figure 11

- Please also report standard deviations in Table 5, Table 6, Table 7

**Strengths And Weaknesses:**

== Strengths ==

This paper is generally well-written, and the logical flow is clear. I would say this paper is also well-motivated and proposes an interesting test-time inference algorithm. The resulting method is very simple, and the authors provide some figures and a toy example to show the readers the key idea behind their method, which I personally like very much. It is also good to see that the authors enrich the introduction part and make the motivation of this paper clearer.

The authors combine their method with three off-the-shelf offline RL algorithms, and conduct some experiments on the D4RL locomotion datasets. The authors also conduct experiments on the Humanoid datasets, where the authors collect the corresponding static datasets by themselves. Furthermore, the authors conduct experiments on Robotic Manipulation tasks. One can observe performance improvement by building MCEP upon numerous base algorithms. To summarize, the strengths and the advantages of this manuscript are

- this paper is well-written with a clear logic flow

- the core idea and the resulting method of this paper is quite simple and easy to implement

- the improvements from the proposed method are significant on many base algorithms

- the authors provide source codes, and I believe that the results presented in this paper are reproducible

- the authors consider many tasks and extensively examine the performance of their proposed method

- the limitation part is well-stated and highly appreciated

Overall, I believe this paper is qualified to be published in TMLR

== Weaknesses ==

Some of my concerns in ICLR2024 review are addressed in the current version. My only criticism to this version is the lack of statistical significance, e.g., error bar and standard deviation. **This applies to almost all of the tables and figures presented in the paper**.

---

> ### Author Response · Authors · 2024-02-14
> **Reply to Reviewer 5tm2**
>
> We appreciated your comments from ICLR and the new suggestions.
>
> ### Response to Requested Changes
>
> In the latest revision:
>
> - We added error bars in Figure 7, Figure 8, Figure 11
>
> - We reported standard errors in Table 5, Table 6, Table 7
>
>
> Again, thank you for your practical suggestions that help bring our paper to its current version.

---

> > ### Comment · Reviewer_5tm2 · 2024-02-17
> >
> > Thanks for addressing my comments. I am satisfied with the current version.

---

### Decision · Action_Editor_buaN · 2024-05-15

**Recommendation:** Accept as is

**Comment:**

The authors have engaged with the reviewers and made changes to the paper based on their suggestions. The review discussion pointed out that the authors can further improve the manuscript and have the following suggestion:
*  The technical motivation could be written more clearly and some of the terminologies can be used with greater consistency.

I'm recommending acceptance, but strongly encourage the authors to consider making the above minor writing improvements for camera ready.

**Audience:**

This paper will significantly inform the offline RL and broader RL community on better methods for action selection at policy inference time.

**Claims And Evidence:**

This paper presents a method for improved test time inference for offline RL. The reviewers found the paper to be generally well-written and the claims are backed up with several experiments on various benchmarks. The authors also release code which will be valuable for reproducibility.